


# Polar substorm on 07 December 2015: pre-onset phenomena and features of auroral breakup

Vladimir  V. Safargaleev[1], Alexander  E. Kozlovsky[2], Valery M. Mitrofanov[1]

[1]Polar Geophysical Institute, Apatity, Russia

[2]Sodankylä Geophysical Observatory, Sodankylä, Finland

*Correspondence to*: Vladimir Safargaleev V. (vladimir.safargaleev@pgia.ru)

**Abstract.** Comprehensive analysis of a moderate 600-nT substorm was performed with using simultaneous optical observations inside the auroral oval and in the polar cap, combined with data from satellites, radars, and ground magnetometers. The onset took place near the poleward boundary of the auroral oval that is not typical for classical

substorms. The substorm onset was preceded by two negative excursions of the IMF Bz component with 15-min interval between them, two enhancements of the antisunward convection in the polar cap with the same repetition period, and 15-minute oscillations in geomagnetic H-component in the auroral zone. The distribution of the pulsation intensity along meridian has two local maxima - at equatorial and poleward boundaries of the auroral oval where pulsations occurred in the out-of-phase mode resembling the field-line resonance. At initial stage, the auroral breakup

developed as auroral torch stretching and expanding poleward along the meridian. Some later it took a form of the large-scale coiling structure that also distinguishes the considered substorm from classical one. Magnetic, radar and AMPERE satellite data show that before the collapse the coiling structure was located between two field-aligned currents: downward at poleward boundary of structure and upward at equatorial boundary. The set of GEOTAIL satellite and ground data fits to the near-tail current disruption scenario of the substorm onset. We suggest that the 15-

min oscillations might play a role in the substorm initiation.

## 1 Introduction

### 1.1 Location of substorm onsets as inferred from satellite and ground observations

Although the substorm onset and development mechanisms were of high interest for many decades, there are still a number of issues under discussion. The substorm studies use satellite plasma and fields measurements in the

magnetotail plasma sheet and simultaneous auroral and magnetic observations on ground in the auroral zone where the plasma sheet is mapped onto the ionosphere. One of the longstanding problems is where and when key substorm processes initiate. The question is solving on the base of satellite observations, whereas direct association of ground disturbances (magnetic variations, auroras etc) with the processes in the satellite vicinity is difficult because of low accuracy of the mapping between distant magnetotail and the ionosphere. Thus, one needs either appropriate

modification of the geomagnetic field model (Brito and Morley, 2017) or involving some additional information (e.g. Shevchenko et al., 2010) to perform more or less accurate conjugation of the satellite with ground instruments.

Two competing substorm scenarios based on in-space observations have been proposed. The first one implies that substorm originates in the near-Earth portion of the plasma sheet due to the dawn-to-dusk current disruption (CD) around 10 $R_E$ in the course of development of some kind of MHD or kinetic instability (e.g. Lui, 1996). In particular, the

ballooning instability (e.g. Roux et al., 1991) may cause current disruption in a localized region of plasma sheet. As a result, the current wedge is formed, auroral structure in the form of westward traveling surge develops and the magnetic

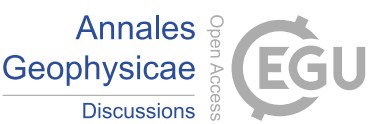

field is dipolarized. In accordance with the second scenario (e.g. Baker et al., 1996), the substorm starts at 20 –30 $R_E$ as a result of magnetic reconnection via near-Earth neutral line (NENL) formation. In ionospheric projection, the closer substorms are associated with maximal geomagnetic disturbances (negative bays in H-component) deep inside the auroral zone whereas distant substorms should be displayed as negative bays with maximum amplitude at higher latitudes (close to the poleward boundary of auroral zone).

Two types of the ground substorm onsets that map into the inner and mid tail were described by Baker et al. (1993) and Pulkinen et al. (1998). In the interpretation of the authors, both types of onsets are initiated by NENL formation. Another point of view is that both CD and reconnection may operate producing different types of substorm onsets in two different latitudinal zones on the ground (Vasyliunas, 1998). Kleimenova et al. (2012) proposed to distinguish the substorms associated with magnetic bays near the poleward boundary of auroral oval ("polar" substorms) from those that start inside the auroral zone and then expand poleward (further referred as "classical" substorms). The statistics show that polar substorms are observed preferentially in the pre-midnight and, indeed, 20% of substorms may be classified as "polar" (Kleimenova et al., 2012). Similar to the classical substorm, the polar substorm is accompanied by Pi2 geomagnetic pulsations and auroral breakup. However, the latter occurs as a large-scale vortex (Kleimenova et al., 2012) or poleward progressing auroral torch-like structure (Safargaleev et al., 2018) rather than an auroral bulge or WTS in the classical substorm onset.

Sometimes substorms occur as a sequence when a clear growth phase is followed by the first onset at lower latitudes and the second one involves all latitudes between 60° and 70° (e.g. Mishin et al., 2001). In the case presented by Safargaleev et al. (2018), the intense polar substorm developed on the "background" of rather weak substorm-like disturbances at lower latitudes. Disturbances started 15-20 min prior the polar substorm onset and may be identified in the westward electrojet.

### 1.2 Substorm triggers

Multiple onsets occur often. If they occur before the main breakup, they are called pseudobreakups (e.g. Koskinen et al., 1993). After the main onset they are called "intensifications". Pseudobreakups look similar to substorm expansion but are relatively weaker. Some researchers believe that pseudobreakups may be regarded as a substorm trigger (e.g. Rostoker, 1968).

The substorm trigger in the interplanetary medium is one more discussion issue. Substorm may be initiated by variations in solar wind dynamic pressure (sudden impulses, SI) or interplanetary magnetic field (IMF). It was found that majority of SI events do not lead to substorms (Liou et al., 2017 and reference therein). Variations in the IMF Bz component seem to be more effective. Russel (2000) suggested that double storm onsets can be caused by a temporal deflection of northward IMF to southward. Mishin et al. (2001) and Safargaleev et al. (2018) proposed that the polar substorm might be initiated by the quasi-sinusoidal variation in IMF Bz component with period ~ 15 min detected in the solar wind several tens minutes prior onset. However, to associate substorm onset with a certain IMF variation one needs careful estimating of the time delay between the arrival of IMF irregularity to the magnetopause and the beginning of the substorm.

The magnetospheric response time to the variation in the solar wind can vary from a few minutes to several hours. Hairston and Heelis (1995) observed a time lag of 17–25 min in the ionospheric flows responding to the IMF changing from north- to southward. In accordance with the numerical simulation of Bargatze et al. (1999), the substorm occurs





30 - 60 min after the solar wind energy input (i.e. after a southward turning of the IMF and dayside reconnection
beginning). This means that time lag between the convection response and the substorm onset might be about 30 min.
One more important but uncertain (within 5-25 min) parameter is the propagation time of solar wind between the bow
shock and dayside magnetopause. Samsonov et al. (2017) showed that the typical time for a southward interplanetary
magnetic field turning to propagate across the magnetosheath is 14 min.

**1.3 Pre-onset phenomena**

Auroral activity at high latitudes contains information about magnetospheric processes. For this reason, a number of
optical studies were focused on the magnetospheric phenomena prior the substorms aiming to find out the precursors of
substorms. Pellinen and Heikila (1978) and Baumjohann et al. (1981) showed that breakup is preceded by the pre-
existing arc fading after its short brightening.  Safargaleev and Osipenko (2001) noted that fading/brightening of
multiple pre-existing arcs looks like poleward displacement of the auroral activity, which may be considered as an
ionospheric trace of the waves propagating tailward in the plasma sheet. Much attention was paid to the nearly north-
south aligned auroral structures originating at the poleward auroral boundary and progressing to lower latitudes, which
were considered as substorm precursors (e.g. Rostoker et al., 1987). Golovchanskaya et al. (2015) focused on the wave-
like signatures of the east-west type auroral activities which appear before breakup and may be related to ballooning
waves propagating in the plasma sheet. In fact, any form of optical pre-substorm activity could be considered as a
precursor of the onset so that such investigations need continuation to clarify the situation.

**1.4 Objectives of the study**

The main aim of the present paper is a detailed multi-instrumental investigation of a case of polar substorm on 07
December 2015.

First, we describe the main features of the polar substorm inferred from ground observations to show that the most
intense onset begins near the poleward boundary of auroral oval so that the preceding onset-like features at lower
latitudes look like pseudobreakup events (section 3.1). In order to avoid discussing whether they are pseudobreakups or
not, we use in the text a general term "pre-onset phenomena".
Second, we show signatures of pre-onset phenomena in the ionospheric radar data (section 3.2) and in the solar wind
(section 3.3).
Third, we emphasize the differences between polar and classical substorms in the auroral data (section 4.1) and
distribution of large-scale field-aligned currents (section 5.2).
Fourth, we present GEOTAIL satellite data to investigate what process in plasma sheet – current disruption or neutral
line formation – is responsible for the substorm onset (section 4.2).
Fifth, we discuss the possible role of 15-min oscillations in the IMF, ionospheric plasma flow, magnetic and optical data
in substorm process (section 5.3)

Finally, we discuss possible mechanisms matching the observations (section 5.4).

**2 Instrumentation**

The study utilizes data from the IMAGE magnetometer network (Tanskanen, 2009). Small black circles in the map in
Figure 1 show location of the magnetometers. Time resolution of the data is 10 s. The time of substorm onset was
defined as the beginning of negative deviation in H-component first detected at Bear Island (BJN, 74.50° N, 19.20° E)
at $T_0 \sim 17:30$ UT. In addition to the magnetograms, we used data of the ionospheric equivalent currents provided in





frame of the ECLAT project (Amm and Viljanen, 1999; Pulkkinen et al., 2003). The equivalent currents are "virtual" currents in the ionospheric plane causing the same magnetic field change on the ground as the real three-dimensional ionospheric/magnetospheric current system. In the equivalent current map, footprints of localized downward (upward) field-aligned current (FAC) are manifested by quasi-circular clockwise (counterclockwise) equivalent current vortices around location of the upward (downward) FAC (e.g. Palin et al., 2016).

Two all-sky cameras (ASC) located in Barentsburg (BAB, 78.09° N, 14.21° E) and Sodankylä (SOD, 67.37° N, 26.63° E) monitored auroral activity. The BAB camera was operating in visible light and provides 1 frame per a second. Green line images from SOD camera at 3 – 10 s resolution were used in the study. Large circles in Figure 1 show fields of view of the cameras at a height of 110 km for elevation angles above 15°. The ASC keograms in Fig. 2 were made along the geomagnetic meridian.

The WIND satellite and two satellites of THEMIS mission (Time History of Events and Macroscale Interactions during Substorms, THB and THC) provided the IMF and solar wind data. This allowed us to estimate interplanetary conditions at the bow shock. The GEOTAIL satellite monitored dawnside plasma sheet parameters and was magnetically conjugated to the region of ground-based observations. DMSP F18 measurement of precipitating particles twenty minutes before the onset allowed us to estimate the location of BJN station as to be close to the poleward boundary of auroral oval. Data of AMPERE satellite (Active Magnetosphere and Planetary Electrodynamics Response Experiment) were used to support conclusion regarding field-aligned current distribution in the area of optical observations.

The European Incoherent Scatter Radar on Svalbard (ESR) is located near Longerbyen (LYR, 78.2° N, 15.8° E) that is about 40 km east of the BAB ASC. The ESR provided height profile of ionospheric parameters (electron density, electron and ion temperatures, and the ion line-of-sight velocity) at 1 min resolution. Data from the Super Dual Auroral Radar Network (SuperDARN) were used for monitoring the ionosperic plasma flow. At the F-region heights, the Doppler shift of received signals gives the line-of-sight component of the convection velocity. A detail description of the system was given by Greenwald et al. (1995) and Chisham et al (2007).

### 3 Pre-onset activity

#### 3.1 General overview of magnetic and auroral activity

The event was during a moderate geomagnetic activity ($Dst \sim$ - 10 nT, $Kp \sim$ 2+). No a magnetic storm occurred a week before and after the event. Variations of geomagnetic H-component at IMAGE stations, the auroral activity above Northern Scandinavia and Spitsbergen as well as the equivalent ionospheric currents (electrojets) are shown in Fig.2. Substorm started at $T_0 \sim$ 17:30 UT ($\sim$ 19:30 MLT) as a strong negative deviation of about $\sim$ -600 nT first seen at BJN (Fig.2a, middle panel) and poleward displacement of the westward electrojet in Fig.2b (top panel). A few minutes later a positive bay with amplitude $\sim$+250 nT was detected at KIL and SOD. As well, noticeable positive variations were seen at mid-and low-latitude stations NUR (Fig.2a) and ABG (see section 4.2), respectively. While negative variations in H-component should be caused by a change in the westward ionospheric current, positive deflections at subauroral latitudes indicate the ionospheric current of opposite direction over SOD. Indeed, both currents are seen in Fig 2b.

The auroral spatial distribution is presented by the keograms in Fig.2a,c. No distinct auroras were seen within field of view of BAB all-sky camera until the onset. Most likely, BAB was in the polar cap at that time. The prevailing auroras over SOD were diffuse auroras which equatorial edge moved from zenith toward the south horizon from 17:00 UT till the moment $T_0$. This means that just before the breakup SOD was inside the auroral oval close to its equatorial



boundary. The position of IMAGE stations relatively the poleward boundary of auroral oval may be estimated from the DMSP F18 data under assumption that the boundary is oriented along the geomagnetic latitude (we use the Altitude Adjusted Corrected Geomagnetic, AACGM, coordinates). The ionospheric projection of the DMSP trajectory 20 minutes before the substorm onset is shown in Fig.3a. In accordance with Newell et al. (1996), the poleward boundary

of the main auroral oval is determined as an abrupt drop in the electron energy flux (*b5e*-boundary in Fig.3b). In Fig.3a the footprint of this boundary is marked by the yellow asterix. Its geomagnetic latitude is 71.4°N that is slightly poleward BJN (71.2°N). Thus, by the moment $T_0$ the BJN was located inside the auroral oval in the vicinity of its poleward boundary. Following Kleimenova et al. (2012), the event can be considered as a polar substorm. Note, that the boundary of diffuse aurora which is well seen in Fig.3a may be associated with ion isotropic boundary (*b2i* – boundary

on DMSP spectrogram). In Fig.3a the footprint of this boundary is marked by the open asterix.

Auroral breakup started at about $T_0$ as one-minute fading and then brightening of the pre-existing auroral arc observed by SOD all-sky camera at zenith angle ~ +75°, i.e. about 400 km north of Sodankylä. Such a behavior of auroras is typical for beginning of a substorm (e.g. Pellinen and Heikkila, 1978). At about same time, active auroras appeared on the southern horizon of BAB ASC, more than 600 km south of Barentsburg. These auroras are better seen on the upper

keogram in Fig.2c from 17:31:30 UT. Although both cameras observed enhanced luminosity somewhere in a vicinity of BJN, because of the large zenith angles we cannot say for sure whether this is the same arc. In the course of breakup development, poleward boundary of auroras in BAB continued the poleward movement whereas equatorial edge of discreet auroras in SOD moved in opposite direction (Fig.2c).

**3.2 Pre-onset phenomena in the data of ground-based observations**

The substorm was preceded by two negative variations in the H-component at KIL and SOD with repetition period of about 15 minutes (see Fig.2a middle panel). Hereinafter, the repetition period means the interval between two consecutive extremes (maxima or minima). The variations were associated with equatorward expansion/displacement and enhancement of the westward electrojet (Fig.2b). At this time the westward electrojet was about three times stronger than the eastward electrojet. As well, two enhancements and slight poleward displacement of discrete auroras

(arc1 and arc 2) took place in SOD (Fig.2a, bottom keogram). The enhancements started at nearly same time as the negative variations in SOD, at 17:13 UT and 17:24 UT, respectively. These moments are shown on the keogram by white arrows. Presumable location of the arc 1 with respect to the electrojets at 17:15 UT is shown in Fig.2b by black rectangle. The features listed above might indicate a pseudo-breakup, however we will use below the term "pre-breakup phenomena" instead of pseudo-breakup.

Six SuperDARN diagrams in Fig.4 show signatures of the large-scale ionospheric plasma flow. As it was mentioned in Introduction, the time lag between convection response and substorm onset might be about 30 min. In such a case, one should look for related convection feature a half hour before $T_0$, i.e. around 17:00 UT. Probably such a feature is the enhancement of the plasma flow in polar cap started at 17:04 UT, reached maximum at 17:08-17:10 UT (diagram *d* in Fig.4) and lasted until $T_0$. One more flow enhancement took place at 16:52 UT, i.e. 15 minutes before the first one

(diagram *b* in Fig. 4). We suggest that the time lag and the close repetition period (~15 min) indicate a relationship of the flow enhancements and the magnetic and optical pre-breakup events.

The first flow enhancement was observed near noon at 78°-85° GLAT (diagram *b* in Fig. 4). This location corresponds to the ionospheric projection of the mantle (Newell and Meng, 1992). So that, the increase of antisunward convection might be caused by the enhancement of the dayside reconnection under negative IMF Bz.



Just before $T_0$ one of the SuperDARN radars detected the enhancement of convective stream toward Spitsbergen (Fig.4f). In Fig.5 we present altitude profiles of the electron density and ion temperature over Spitsbergen measured by ESR, where time $T_0$ is indicated by a white arrow. The increase of F-region electron density at about $T_0$ looks like a signature of the polar patch associated with the reconnected flux tubes drifting across the polar cap from the cusp to the magnetotail (e.g. Lockwood and Carlson, 1992). Appearance of the polar patch corresponds to the equatorward shift of

westward electojet (Fig.2b). No auroras were found in the BAB optical data to be associated with this pre-onset patch.

**3.3 Pre-onset phenomena in the interplanetary space**

Positions of the satellites measuring interplanetary parameters (THB, THC and WIND) are shown in Fig.6a. The satellites coordinates and the bow shock and magnetopause locations were obtained via the Interactive visualization of satellite orbits tool (4-D Orbit Viewer) available from CDAWEB system. From the THEMIS satellite data we have

obtained about 650 km/s propagation velocity of the IMF features indicated by shadow in Fig.6b. This corresponds to the solar wind speed measured at the WIND satellite. Assuming the nose of the bow shock at 14 $R_E$, we get the propagation time from THC to the bow shock about 6 minutes. The propagation time through the magnetosheath can be estimated as 14 min (Samsonov et al., 2017). Thus, the southward turning of IMF Bz could reach the magnetopause 20 min after registration onboard THC, and the ionospheric convection is expected to respond in ~ 20 min after that

(Hairson and Heelis, 1995).

Shadow areas in Fig.6b indicate the IMF Bz feature which shape and time well correspond to the features of ionospheric convection discussed above. Indeed, there are two southward IMF deflections through 15 minutes, and the first deflection was detected at THC 40 minutes before the first flow enhancement in the polar cap (diagram *b* in Fig. 4). At the moments 16:15 UT and 16:30 UT when Bz at THB reached its maximal (negative) values, the IMF By

component was near zero. This was favourable for reconnection at the subsolar magnetopause. Importantly, the solar wind dynamic pressure does not show essential variations during the interval (top panel in Fig.6b). We use this fact in section 4.2 to exclude the influence of solar wind on the magnetic field variations near equator.

**4  Features of the polar substorm onset**

**4.1 Auroral breakup**

As was mentioned in Section 3.1, the auroral breakup started at about $T_0$ as the brightening and poleward displacement of the most equatorial auroral arc located slightly poleward of the northern coast of Scandinavia. The arc was too far away from zenith of SOD for correct mapping. In the lack of optical observations between SOD and BAB, we can only suppose that the arc was between the westward and eastward electrojets and moved poleward together with them. Presumable location of the arc is shown by black rectangle in Fig.2b. Thus, for the first few minutes the auroral activity

developed according to the traditional scenario.

Auroral situation has changed at ~ 17:38 UT when amplitude of the negative H-component variation at BJN reached a maximum and a more rapid decrease of the H-component at LYR began (moment $T_1$ in Fig.2a). Keograms in Fig.2c show that after this moment the auroras within the field of view of BAB and SOD cameras moved in opposite directions. The auroras seen in SOD expanded almost 600 km equatorward, while the auroras observed in BAB shifted

about 1000 km poleward. So that, by 17:42:37 UT the auroral configuration resembled the double-oval structure of 1600 km in width with bright poleward and equatorial edges and rather weak auroras inside. The next poleward excursion of auroras in BAB with less prominent equatorward shift in SOD started at 17:49:32 UT and reached the





northern horizon in the BAB camera field of view at 17:57:46 UT. The interval between the maximal expansions of auroras to the north was about 15 minutes, which is about the same as, first, the repetition period of variation in geomagnetic H-component in SOD and KIL (Fig.2a), second, the interval between the two negative excursions of IMF Bz-component (Fig.6b), and third, the interval between the two bursts of antisunward flow in polar cap (Fig.4).

Poleward displacement of the auroras started at about $T_1$ as appearance of a new arc closer to BAB zenith than pre-existing auroras (Fig.7a, image at 17:38:03UT). The new arc included a series of bright patches. This feature is often referred to as "beading" (e.g. Keiling et al., 2012). At 17:38:27 UT one of the patches gave rise to the auroral structure (indicated by thin white arrow in Fig.7a), which looks like an auroral torch (e.g. Tagirov, 1993). At this moment the structure was stretched approximately along geomagnetic meridian and had dimension of 170x170 km. Then the structure expanded to the west and north, transformed into the large-scale coiling structure (the term was suggested by Akasofu and Kimball, 1964) and broke up into bright strips, rays, patches and vortices at 17:40 UT. The velocity of structure expansion in the first ten seconds was about 5 km/s to North and 10 km/s to East. The auroral distribution before collapsing of the coiling structure is presented in Fig.7b together with the 2D-configuration of the ionosperic equivalent currents.

Two vortices are seen in the current distribution. Center of the first (larger) vortex indicating an upward FAC is located between SOD and BJN. The second (smaller) vortex indicating a downward FAC is located poleward of LYR. Comparison with the auroral distribution shows that the center of the second vortex was poleward of the expanding coiling structure. At ~ 17:39 UT the structure reached the ESR in LYR. This moment is identified in the ESR data as a sharp increase of the E-region electron density (Fig. 5, top panel), which is a signature of auroral precipitation. One minute earlier the ESR detected the ion temperature increase (Fig.5, bottom panel), which indicates enhanced electric field just poleward of the auroras.

To summarize, the downward field-aligned current was detected at the poleward side of coiling structure and there was an upward field-aligned current equatorward of it.

### 4.2 Signatures of disruption of dawn-to-dusk plasma sheet current

During the event, the GEOTAIL satellite was in the near equatorial magnetotail at 16 $R_E$ and ~ 18 LT (Fig.6a). The satellite footprint was calculated using the 4-D Orbit Viewer (see section 3.3). With taking into account the results of (Safargaleev and Safargaleeva, 2018) on the accuracy of distant satellite mapping, the latitude of GEOTAIL footprint was estimated at 75 ± 3° N. The footprint is shown in Fig.7b (left panel) by black square. At the moment indicated in the 2-D diagram, the GEOTAIL position was mapped to the region of the westward electrojet.

Figure 8a shows magnitude of the magnetic field at the GEOTAIL location. Before the onset at 17:30 UT the horizontal Bx component drastically exceeded Bz component, which means that satellite was near the neutral current sheet (the cross-tail current is directed from dawn to dusk). After the time $T_0$ GEOTAIL was measuring gradual decrease of the differential flux of energetic ions accompanied by the decrease in Bx component (indicated by gray shadow) while Bz component almost did not change. At this time the westward electrojet where GEOTAIL was mapped has been enhanced (Fig. 2b). These features of the magnetic filed, particle flux and westward electrojet indicate a decrease or even local disruption of the dawn-to-dusk current in the vicinity of GEOTAIL. The local disruption of the cross-tail current causes partial diversion of the current into the ionosphere and formation of the substorm current wedge.



The current disruption is also supported by the positive variation in the H magnetic field component at the low-latitude station Alibag (ABG, 18.5°N, 72.9°E) located near midnight (Fig.8b, bottom panel). The increase of H-component at low latitudes is traditionally connected with the enhancement of solar wind dynamic pressure which is not in a present case (see section 3.3). In accordance with Maltsev et al. (1996) and Huang et al. (2004), the cross-tail magnetospheric current also contributes to the Dst variation, i.e., it decreases the H-component at equatorial latitudes. Hence, the

magnetic effect of the current disruption is manifested as the H-component increase at the equatorial latitudes. Such an effect was not seen on the dayside (e.g. at station San Juan, SJG, 18.1°N, 293.8°E in Fig.8b), which indicates the current disruption in the magnetotail (on the night side only).

The spectrogram from GEOTAIL (Fig. 8a top panel) shows that at 17:55 UT flux and energy of protons start to increase. This was accompanied by the Bx reduction and Bz increase that indicates dipolarization of magnetic field at

the GEOTAIL location. Five minutes later the increase of the flux stopped. Figure 2a shows a secondary weaker onset at BJN at this moment, whereas at the higher latitudes (LYR) the recovery phase started. This is different from the case described by Baker et al. (1996) who observed that the recovery phase started in auroral zone and a new negative bay started at higher latitudes (i.e., on opposite to our case). Assuming that the reappearance of the energetic ions in Fig.8a indicates rapid plasma sheet thickening (Baker et al., 1996), one can suppose that the dipolarization and second

onset/intensification were due to the neutral line formation.

## 5 Discussion

### 5.1 Summary of pre-breakup observations

We identify the substorm onset time, $T_0$, as beginning of the negative bay at the high latitude station BJN. As well, at this time the intensification and poleward displacement of the westward electrojet began (Fig.2b). The auroral breakup

started around $T_0$ as one minute fading and then brightening of the pre-existing auroral arc at about 400 km north of Sodankylä. The DMSP data of precipitating particles show that 20 minutes before $T_0$ the poleward edge of the auroral oval (*b5e*-boundary in Fig.3b) was near BJN. For this reason, following Kleimenova et al. (2012), we attributed the event to the subclass "polar substorms".

The polar substorm was preceded by two rather weaker (~ 80 nT) negative bays, recorded by IMAGE magnetometers

deep inside the auroral oval and following each other through a 15 min interval. The bays were accompanied by brightening of the auroras near the north edge of SOD camera field of view and their poleward displacement. Pre-onset phenomena of the same periodicity were found in the polar cap plasma flow and IMF variations.

The search for pre-onset phenomena in the ionospheric convection and in the solar wind was based, firstly, on the time response of the magnetosphere to solar wind changes and, secondly, on the observation of the 15-minutes periodicity.

The search results are shown in Figs.6 and 4b and represent two negative excursions in IMF Bz-component and two bursts of the antisunward ionospheric plasma flow across the polar cap, respectively. Earlier Russell (2000) discussed possible role in the "classical" substorm development of a single negative Bz variation (i.e. when the northward IMF turns southward and then northward again). However, Safargaleev et al., (2018) proposed that the polar substorm might be triggered by a quasi-sinusoidal variation in Bz.

The hypothesis of dayside reconnection is supported by the density patch observed by ESR in polar cap at about $T_0$ (see Fig. 5). Accordingly to Lockwood and Carlson (1992), the patch may be associated with the reconnected flux tube moving from cusp to the lobe, and the plasma flow from polar cap to the auroral oval during the substorm pre-onset





phase was observed by Mishin et al. (2017). The patch in the ESR data was associated with a southward displacement of the poleward boundary of the westward electrojet (Fig.2b). Taking into account that BJN and, hence, the westward

electrojet were near the polar cap, the southward shift of the electrojet boundary indicates the "swelling" of magnetotail lobe in the course of energy storage. The swelling of the both lobes leads to plasma sheet thinning that makes it instable due to highly stressed magnetic configuration.

Optical observations in the polar cap near the boundary of the auroral oval do not reveal any aurora which might be attributed to the electron density patch in the ESR data. The lack of optical data over BJN (see Fig.1) do not allow us to

conclude whether the patch was associated with a poleward boundary intensifications (PBIs).

**5.2 Summary of breakup observations**

Auroral breakup at the initial stage proceeded as brightening and poleward displacement of one of pre-existing arcs located deep inside the auroral oval, presumably, between the westward and eastward electrojets near the poleward edge of diffuse auroras seen from SOD. After that smaller-scale (comparing to WTS or auroral bulge) structure has

originated from the bright spot at the south horizon of BAB and expanded westward and poleward at the velocity 10 and 5 km/s, respectively, which is close to a typical velocity of the WTS expansion. During the first few seconds the structure resembled the auroral torch, but before the collapse it had a coiling shape. Akasofu (1977) showed that WTS develops typically at latitudes between 65° and 70° whereas in the present case the torch-like structure appeared higher than 70° N GLAT. Sergeev and Yahnin (1979) observed that the substorm bulge originates equatorward of the open-

closed field line region and then expands up to but not beyond a more poleward arc system which, perhaps, delineates the open-closed field line boundary. In the present case no auroras were seen poleward of the torch formation near the poleward boundary of the auroral oval (*b5e*-boundary in Fig.3b, section 3.1). Hence, the generation mechanisms for torch and WTS may be different.

The moment of generation of the torch-like structure was preceded by formation of series patches along the arc

(beading structure). This structure was regarded by Keiling et al. (2012) as a signature of the interchange instability on the outer boundary of the plasma sheet which might be responsible for the torch appearance. If the *b5e*-boundary corresponds to the ionosheric projection of the outer edge of the plasma sheet, the interchange hypothesis looks reasonable. Earlier Rezhenov (1995) suggested this kind instability to explain generation of the transpolar arc.

The distribution of field-aligned currents in the vicinity of the coiling structure inferred from the AMPERE

measurements (Fig.7c) shows a downward and upward FAC pole- and equator-ward of the structure, respectively, which corresponds to the statistical results of Iijima and Potemra (1978) showing three current sheets (two downward and one upward between them) in the pre-midnight sector. Note that indeed the polar substorms are preferentially observed in this MLT-sector (Kleymenova et al., 2012). Classical substorms start at lower latitudes where the current distribution is opposed to that for high latitudes, i.e., the upward current is north of the stable arc and downward current

is equatorward (Aikio at al., 2002). Thus, a key difference between the polar and classical substorms may be in the position of the breaking auroras relatively the large-scale down- and upward currents.

Typically auroral arcs occur in the regions of large-scale upward field-aligned currents associated with downward fluxes of electrons. However, Kozlovsky et al. (2005) have shown that at magnetospheric plasma boundaries the Kelvin-Helmholtz (K-H) instability may lead to generation of auroral wave-like forms even in the region of a large-

scale downward FAC. At the initial stage of instability development such structures look like a series of auroral spots



resembling the beading structure. Thus, the K-H instability may be responsible for generation of both the torch-like and coiling auroras. Note also that such configuration of the field-aligned currents in vicinity of breakup auroras hinders the development of interchange instability.

The set of satellite and ground observations (section 4.2) allows us to interpret the gap in the flux of hot ions at the
location of GEOTAIL, which started at the moment $T_0$, as a decrease or local disruption of the dawn-to-dusk current in plasma sheet and its partial diversion into the ionosphere in the course of substorm current wedge formation. The signatures of dipolarization were observed on GEOTAIL 25 min later and we associate the dipolarization with reconnection in the magnetotail and the second onset/intensification at BJN. We note unexpected large positive variation in H-component at the nightside equatorial station (Fig.8b) which we explain by the weakening of currents in
the magnetotail (see also Huang et al., 2004).

Finally, we emphasize the 15-min periodicity in the aurora development. The keograms in Fig. 2c show that after moment $T_1$ auroras over BAB and SOD moved in opposite directions giving the impression of periodical "swelling" of magnetotail plasma sheet. We think that the 15 min periodicity in pre-onset and breakup processes is the most intriguing finding and deserves a more detailed discussion.

**5.3 Periodicity in the processes prior and during the polar substorm onset**

The period about 15 min (frequency 1 mHz) corresponds to the IPCL or Ps6 geomagnetic pulsations. The former are typical feature of the dayside cusp (e.g. Troitskaya, 1985). The latter are a subclass of the Pi3 pulsations (Saito, 1978), which are detected in the Y-component and associated with the omega-auroras (e.g. Jorgensen et al., 1999). As well, signatures of nearly 15-min magnetosphere oscillations were found in the modulation of ULF activity (Safargaleev et al.
2002), the DOPE sounder radar data (Wright and Yeoman, 1999), and the GPS TEC variations (Watson et al., 2015). Thus, the role of nearly 15-min oscillations is not limited only to substorms but may be attributed to wider range of magnetospheric processes.

First, the 15-min periodicity as two negative excursions was detected in the variations of IMF Bz (Fig.6b). Then, there were two consecutive enhancements of the antisunward plasma flow in the polar cap (Fig. 4). The time delay between
the flow enhancements and the IMF Bz variations suggests that the former was a consequence of the latter. A similar repetition period was found in the two negative bays of about 80 nT in H-component and the accompanying aurora intensifications (arc 1 and arc 2) inside the auroral zone (Fig.2a). The bays followed the plasma flow enhancements, and time delay indicated their relation to the IMF Bz variations.

The second feature was found in the latitudinal distribution of the intensity of 15-min geomagnetic pulsations. Figure 9a
demonstrates a "wave portrait" of the polar substorm onset in the frequency band 0.8 ÷1.7 mHz (period ΔT = 15 ± 5 min) for some IMAGE stations. Two maxima at SOD and at Hopen Island, HOP (GLAT =72.85°N) are seen in the latitudinal distribution of pulsation amplitude in Fig.9b, where gray area shows position of the auroral oval 20 min before the onset, as it was estimated in section 3.1. The both maxima were at ~ 17:34 UT. By this time, the expanding auroras as well as the westward electrojet might shift noticeably to the north (gray arrows in Fig.2b), so that poleward
boundary of the auroral oval occurred closer to HOP than to BJN, comparing to that during the DMSP flight. A new presumable location of the footprint of the outer edge of plasma sheet is indicated by gray dashed line. The keogram in Fig 2a indicates that the equatorial edge of the auroral oval was southward of SOD at this time.





For a pure Alfven wave, the period of oscillations is defined by propagating time of the wave between conjugated ionospheres and should depend on the length of the magnetic field line (i.e. on the latitude), however we do not observe

such a dependence in the present case. Although the latitudinal separation of the peaks is very large (about 10°), the pulsations have almost the same period along the meridian (Fig. 9a). Moreover, the magnetosphere is inhomogeneous along the meridian and includes at least three different areas – lobes, plasma sheet and a gap between the plasma sheet and plasmasphere. This observation may be explained by the coupling of Alfven and compressional modes excited from outside by periodic negative excursions of IMF Bz.

The third feature is the out-of-phase magnetic variations at SOD and HOP stations where pulsations have local maxima. Figure 9c shows at least five events of phase-shifts by 180° at the interval of about 7- 8 min (half period of pulsations). Two open arrows indicate the pre-onset enhancement of arc1 and arc2. The moment $T_0$ corresponds to the substorm onset (i.e. the beginning of negative declination at BJN in Fig. 2a) which is also accompanied with brightening of the pre-existing aurora arc over SOD. The moment $T_1$ corresponds to beginning of the auroral torch development in Fig.7a,

which was peceeded by the appearance of a new arc in BAB.

Although the out-of-phase oscillations of two neighboring L-shells is a signature of the field line resonance (FLR), the present case is essentially different from FLR. Namely, the 15-min pulsations are detected in the latitudinal range of ~ 20° at least, whereas typical FLR are observed in a narrow latitudinal range of the order of 2° (Walker et al., 1979). Then, period of FLR is typically less than 10 minutes. Note that frequency of some pulsations may be defined not only

by the internal structure/size of the magnetosphere, but also by the frequency of some external driver (e.g. solar wind) and FLR may be excited from outside (e.g. Walker, 2005).

Following Sarafopoulos (2005), we nominate the out-of-phase oscillations in Fig.9 as pseudo-FLR event. Following Lyatsky et al. (1999), we suppose that the out-of-phase variations of two "neighboring" L-shells (which are inner and outer boundaries of plasma sheet in our case) lead to the field-aligned current between the shells which can be

responsible for intensification of pre-existing arc1 and arc 2, as well as the breakup arcs at the moments $T_0$ and $T_1$.

**5.4 Generation mechanism of polar substorm**

In general, the substorm growth phase occurs as a result of an enhanced dayside reconnection rate, usually initiated by a southward turning of the IMF, concurrent with a comparably small nightside reconnection rate (Milan et al., 2007). However, a number of models of substorm triggering based on observations have been suggested (see Rae et al., 2014

and references therein).

The ground data show that the considered event evolved in four stages. (1) Two enhancements of antisolar convection in the polar cap. (2) Two weak negative deviations in the magnetic field H-component inside the auroral oval that were accompanied by aurora enhancement and looked like the pseudo-breakups. (3) Polar substorm as more intensive negative bay at the poleward edge of auroral oval and, finally, (4) intensification (one more onset) approximately at the

same position. We believe that these stages were due to different reasons and played different roles in the substorm development.

The convection enhancements were caused by negative deviations of IMF Bz component and lead to the increase of magnetic energy in the lobes of the magnetosphere. Two weak variations in H-component might be the ground signature of global oscillations of the magnetospheric cavity excited by periodic erosion of the dayside magnetopause in

the course of periodic reconnection (e.g. Agapitov et al., 2009). Amplitude distribution of the oscillations has two





maxima in the vicinity of equatorial and poleward boundaries of the auroral oval where the oscillations occur in out-of-phase mode. We consider these out-of-phase oscillations as, at least, a reason for the auroral arc intensification via the pseudo filed-line resonance excitation.

The set of satellite and ground data fits to the near-tail current disruption scenario of polar substorm. However, the data set does not allow us to specify a reason for the disruption. We suppose that this might happen due to pseudo FLR. The role of typical FLR event (i.e. out-of-phase variations at two "neighboring" L-shells) in the substorm initiation was discussed in many papers (e.g. Samson et al., 1992; Rae et al., 2014 and references therein). The question whether the out-of-phase variations at inner and outer boundaries of the plasma sheet can be launched from outside and lead to the same effects as the FLR is the subject for a separate theoretical investigation that is beyond the scope of this study. One more possible reason for substorm triggering might be the interchange [Roux et al., 1991] or ballooning (Golovchanskaya et al., 2015; Keiling et al., 2012) instabilities which signature in the form of series of bright patches along the auroral arc we observed just before the breakup.

Finally, the fourth stage of polar substorm development, i.e. second onset or "intensification", is associated with the magnetotail reconnection.

**6 Conclusion**

We present the comprehensive description of the moderate "polar" substorm (the term was suggested by Kleymenova et al., 2012) focusing on the multi-instrumental study of pre-onset events in the solar wind, ionosphere and on the ground. The onset took place at pre-midnight near the poleward boundary of the auroral oval that is not typical for classical substorms. We have shown that the auroral breakup developed between two field-aligned currents with downward current poleward the breaking auroras and upward current south of them. This morphological feature distinguishes the polar substorm from classical ones.

The onset was preceded by two negative excursions of IMF Bz component with repetition period ~ 15 min. These variations caused periodic reconnection at the magnetopause. Two enhancements of the antisunward convection in the polar cap and appearance of the ionospheric patch near the polar cap boundary support the reconnection hypothesis. On the one hand, the reconnection leads to the increase of the magnetic energy in the lobes and corresponding thinning of the plasma sheet that creates favorable conditions for substorm initiation. On the other hand, the periodic erosion of the magnetopause excites the global 15-min oscillation of the magnetospheric cavity. The oscillations are observed in the auroral zone. Period of the oscillations does not depend on the latitude which means that the pulsations represent forced oscillations of the magnetosphere cavity. Latitudinal distribution of the oscillations' intensity has maxima near the equatorial and poleward boundaries of the auroral oval where the oscillations occur in the out-of-phase regime resembling the field-line resonance.

The onset was accompanied by disruption of the dawn-to-dusk current in the plasma sheet around (X, Y) ~ (-16, 16) $R_E$ and the current wedge formation. We conclude this from data of the GEOTAIL satellite magnetically conjugated with the area of ground observations, enhancement of the westward electrojet and the large positive variation in H-component at low latitudes. We think that the onset might be initiated by the out-of-phase oscillations in the same way as field-line resonance does (e.g. Rae et al., 2014). One more possible reason for the substorm triggering might be the interchange or ballooning instabilities. External excitation of the out-of-phase oscillations is regarded as the reason for the auroral arc brightening prior and just after onset.



**Author contribution**: VS provided optical data of the PGI all-sky camera in Barentsburg and AK provided optical data
of SOD all-sky camera in Sodankyla. VM processed BAB optical data as well as mapped auroras and DMSP pass. VS
prepared the manuscript with contributions from all co-authors. The authors declare that they have no conflict of
interest.

**Acknowledgements.** We are grateful to FMI/GEO and other institutes that maintain the IMAGE magnetometer
network (http://space.fmi.fi/image/www/index.php?). We acknowledge V. Angelopoulos at UCB, NASA NAS5-02099
for data from THEMIS, S.Kokubun at STELAB Nagoya University, Japan and D. Williams at APL/JHU for data from
GEOTAIL, A. Szabo and K. Ogilvie at NASA/GSFC for data from WIND, as well as CDAWeb for use all these data
(https://cdaweb.sci.gsfc.nasa.gov/index.html/). The DMSP particle detectors were designed by D. Hardy, F. Rich, and
colleagues at AFRL at Hanscom AFB in Boston. The U.S. Air Force has publicly released this data. Most of it was
obtained through WDC-A (NOAA) in Boulder, with generous supplements from AFRL (http://sd-
www.jhuapl.edu/Aurora/spectrogram/index.html). We thank D. Hardy, F. Rich and P. Newell for its use. The Kp and
Dst indices are from the Kyoto World Data Center C-2 in Kyoto, Japan (http://wdc.kugi.kyoto-u.ac.jp/). Some results
rely on the data collected at ABG and SJG observatories. We thank IIG, India and USGS, US for supporting its
operation and INTERMAGNET for promoting high standards of magnetic observatory practice
(http://www.intermagnet.org/index-eng.php#). The authors acknowledge the use of SuperDARN data. SuperDARN is a
collection of radars funded by national scientific funding agencies of Australia, Canada, China, France, Italy, Japan,
Norway, South Africa, UK, and the United States of America. SuperDARN data are available from the SuperDARN
website hosted by Virginia Tech (http://vt.superdarn.org).The authors appreciate the EISCAT Scientific Association for
making the data observed by the ESR freely accessible on the Madrigal website (PI is I. Häggström
https://www.eiscat.se/schedule/schedule.cgi). We thank the AMPERE team and the AMPERE Science Center for
providing the Iridium-derived data products at http://ampere.jhuapl.edu/index.html. BAB and SOD all-sky cameras are
operated by PGI, Russia (PI Roldugin) and SGO, Finland. We thank N. Safargaleeva (PGI) for BAB optical data
selection. VS acknowledges support from the Academy of Finland via grant 316991.





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






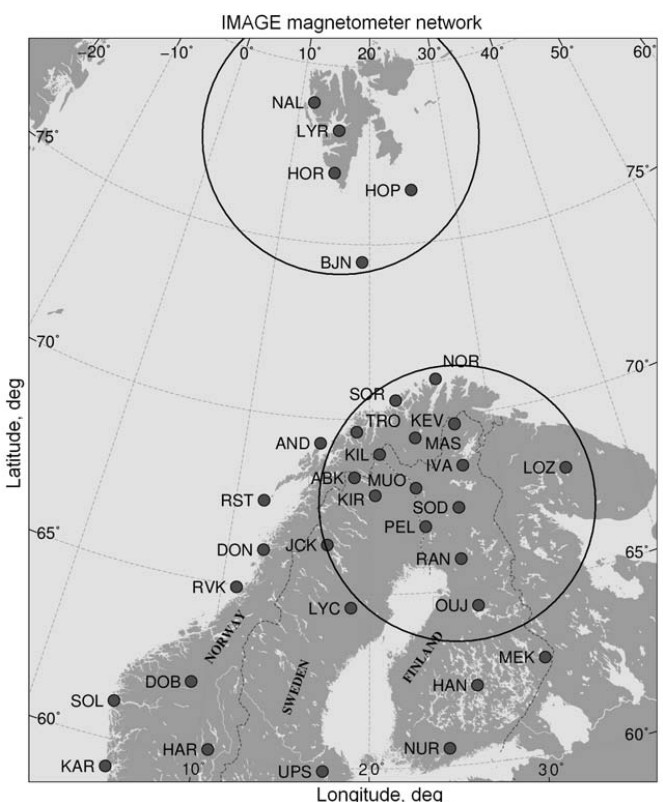

Figure 1. Observatories of IMAGE magnetometer network (small black circles). Large circles show field of view of the all-sky cameras in Barentsburg (*top*) and Sodankylä, SOD (*bottom*).



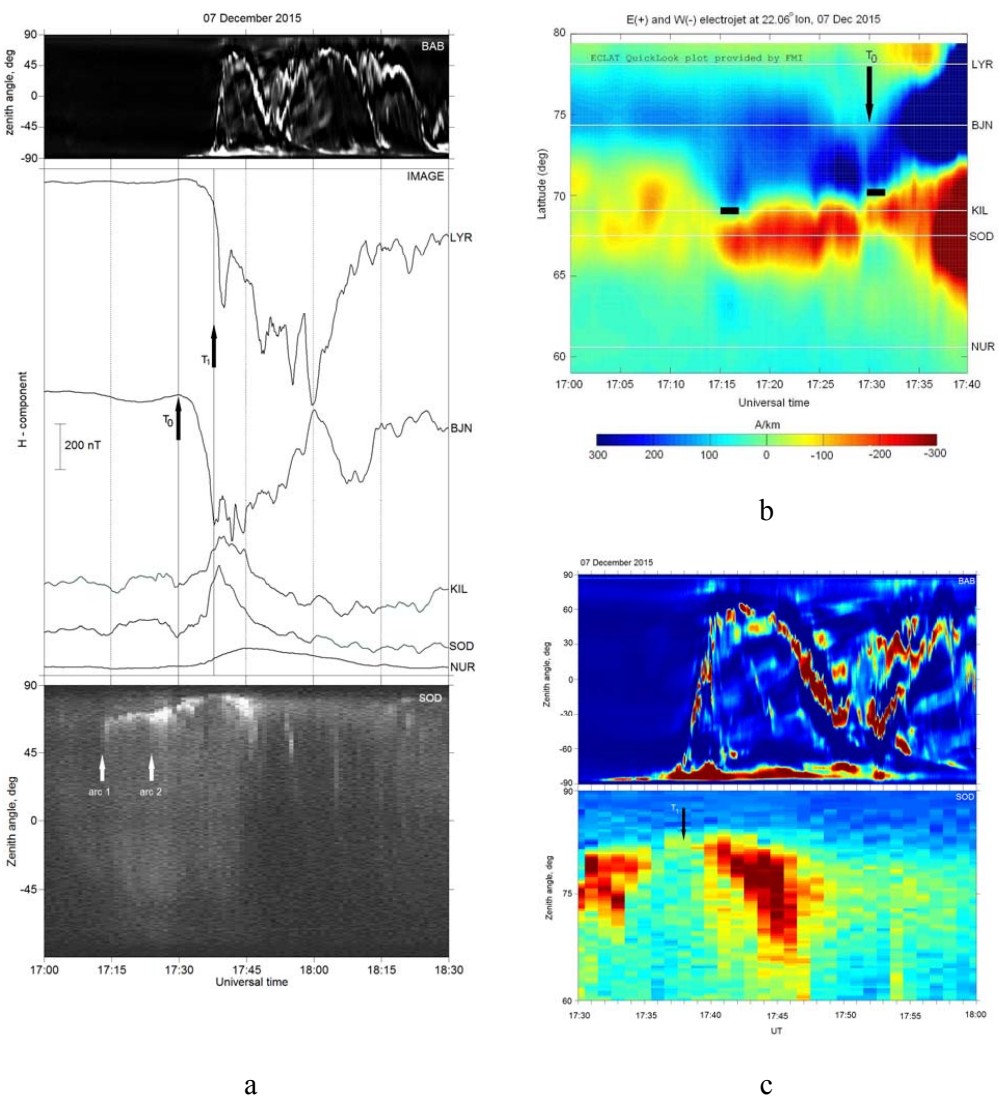

Figure 2. (a) keograms showing aurora dynamics over Barentsburg, BAB (top panel) and Sodankylä, SOD (bottom
panel) and magnetic data from five observatories of the IMAGE magnetometer network (middle panel); (b) dynamics of
equivalent ionospheric currents, westward and eastward electrojets are indicated with gradations of blue and red,
respectively, white horizontal lines show the latitude of the observatories; (c) keograms SOD and BAB at higher
temporal resolution. $T_0$ is the time of polar substorm onset.



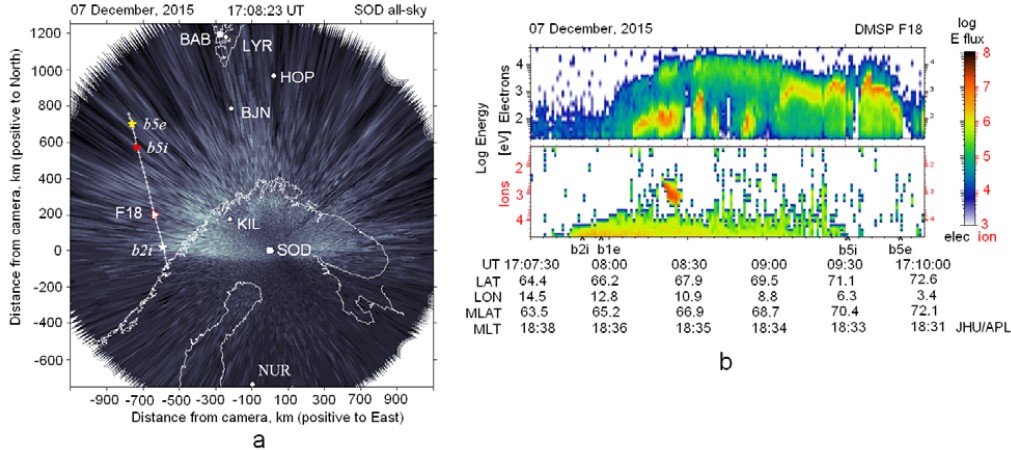

Figure 3. (a) Sodankyla (SOD) all-sky camera image at 557.7 nm. North is up and west is on the left. DMSP F18 trajectory is mapped, and the triangle marks the location of the satellite at the time of the image. (b) DMSP spectrograms with the magnetospheric boundaries identified using algorithms of Newell et al. (1996).





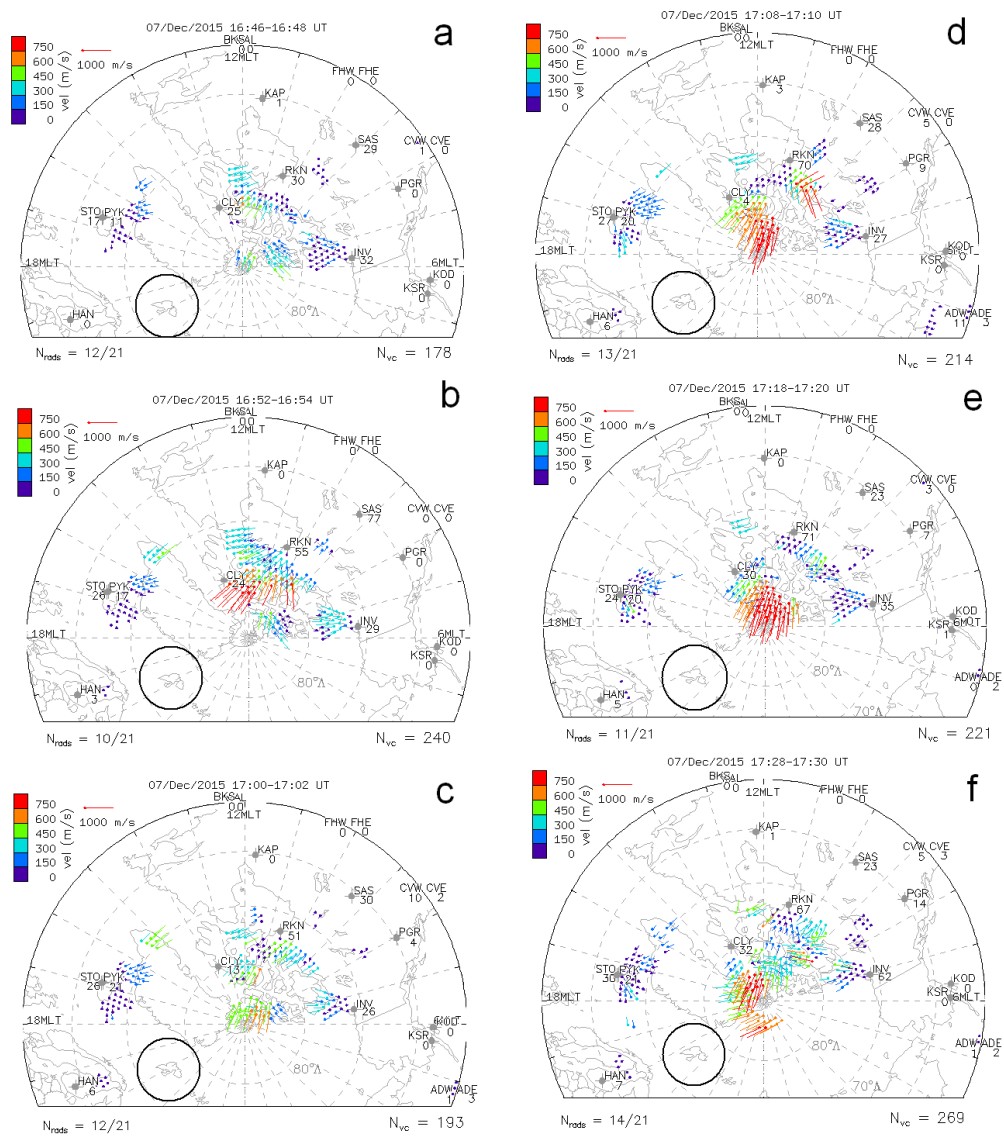

Figure 4. Series diagrams showing global convection patterns averaged over 2 min. Velocity vectors are plotted at the
points where velocity data were provided by measurements. Large circles show field of view of the all-sky camera in
Barentsburg, BAB.



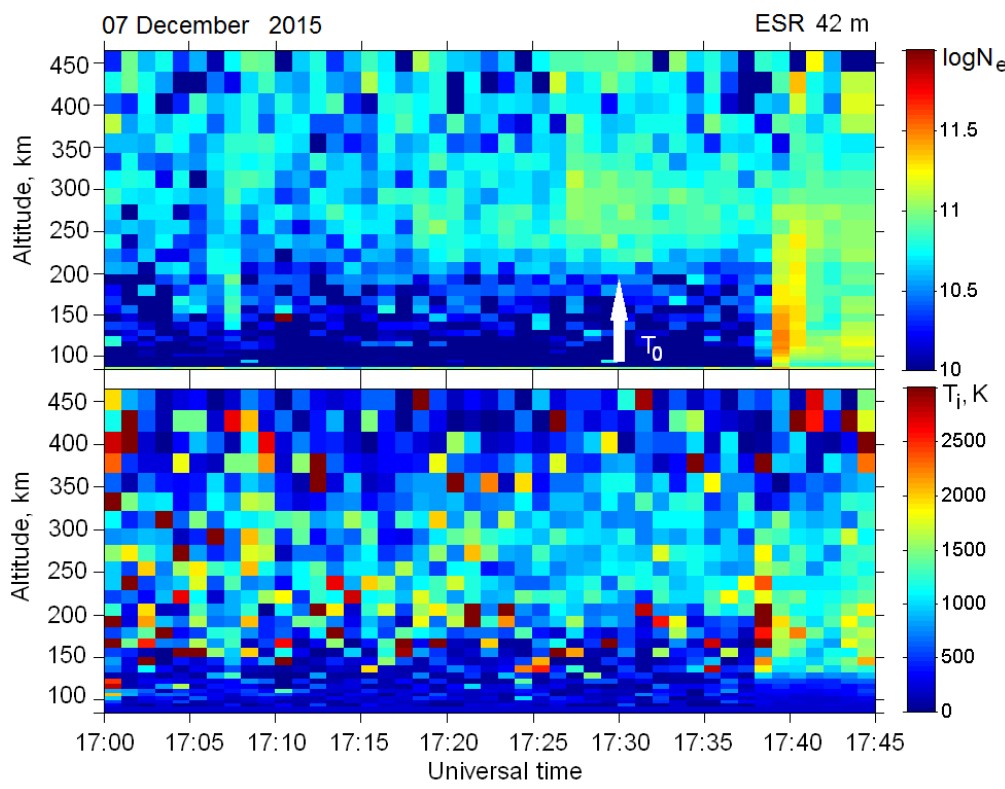

Figure 5. Data of the EISCAT Svalbard radar (ESR) at Longerbyen: electron density $Ne$, and ion temperature $Ti$. $Ne$
enhancement at 17:39 UT was associated with the coiling structure arrives at the beam.





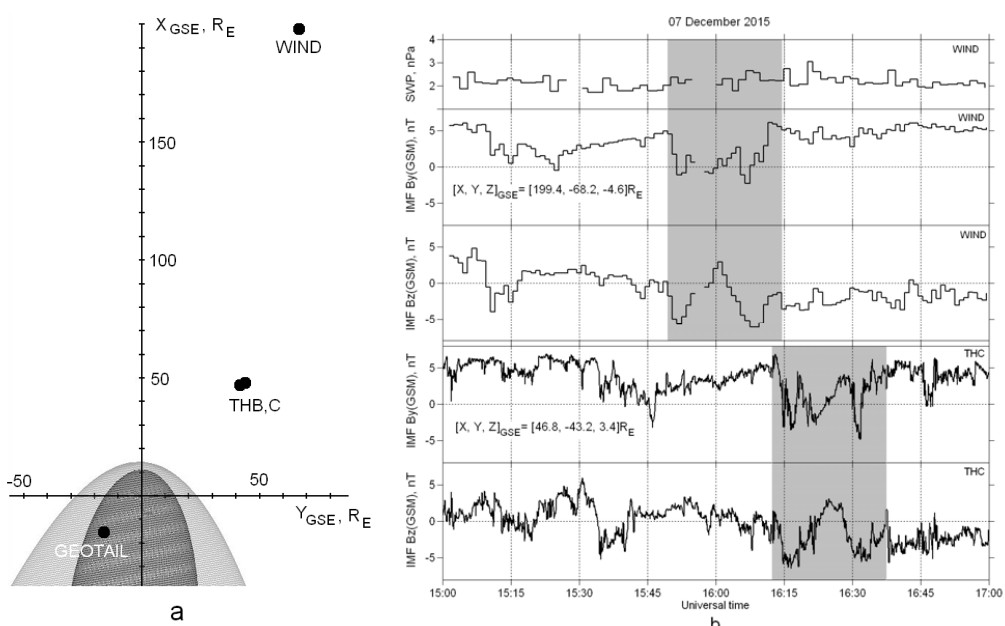

Figure 6. (a) satellite positions in solar wind (WIND, THB and THC) and in the magnetosphere (GEOTAIL); (b) variations of the solar wind pressure and IMF Bz and By components. Two negative excursions of Bz on the both satellites resembling the quasi-sinusoidal variation with period ~ 15 min are highlighted by gray.





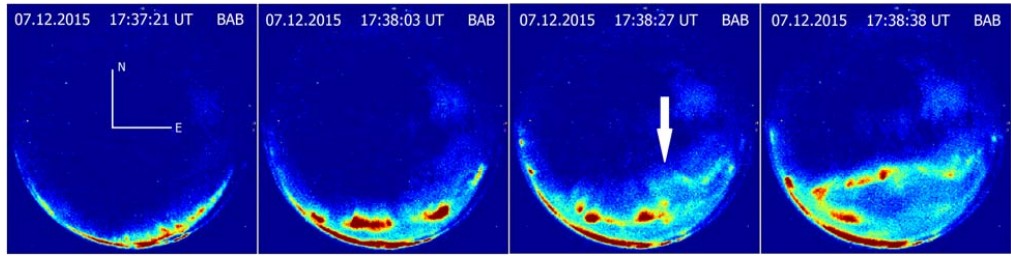


a

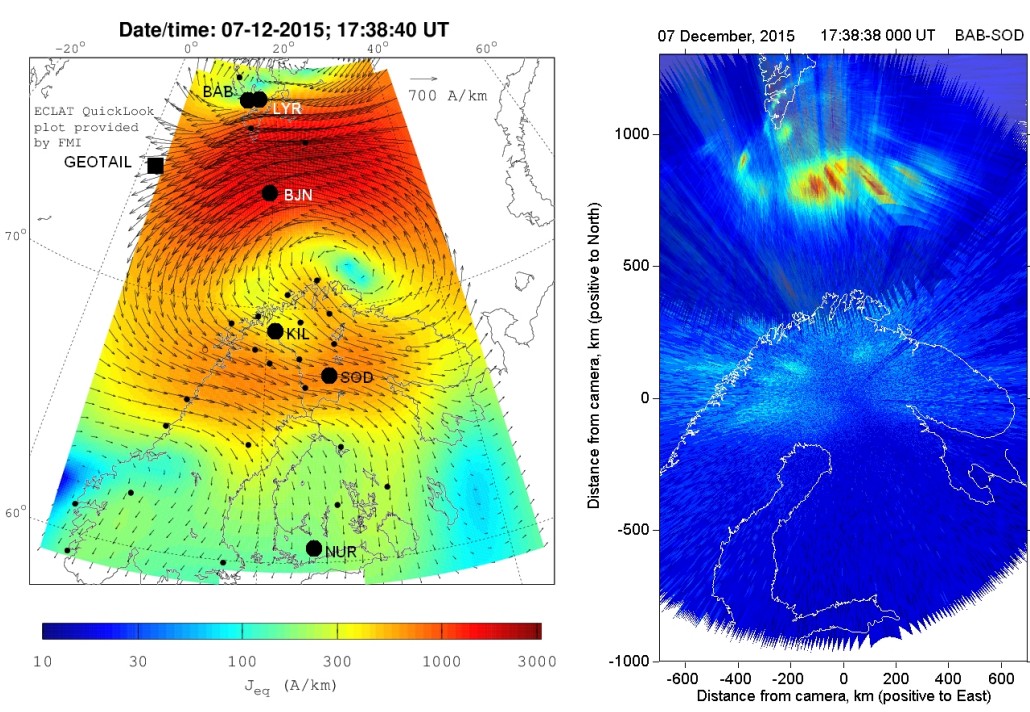

b

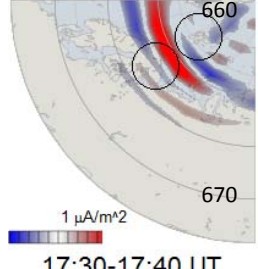

Figure 7 (a) sequence of BAB all-sky images showing the series of bright patches along the enhancing arc and development of the torch-like structure from one of them; (b) *left panel*: snapshot of 2-D equivalent current; *right panel*: mapped SOD and BAB all-sky images, showing the shape of auroras. Black square and circles indicate the position of GEOTAIL footprint and IMAGE observatories, respectively; (c) distribution of the FAC inferred from AMPERE data. Upward currents are shown by red and downward currents in blue. Circles indicate field of view of the all-sky cameras.





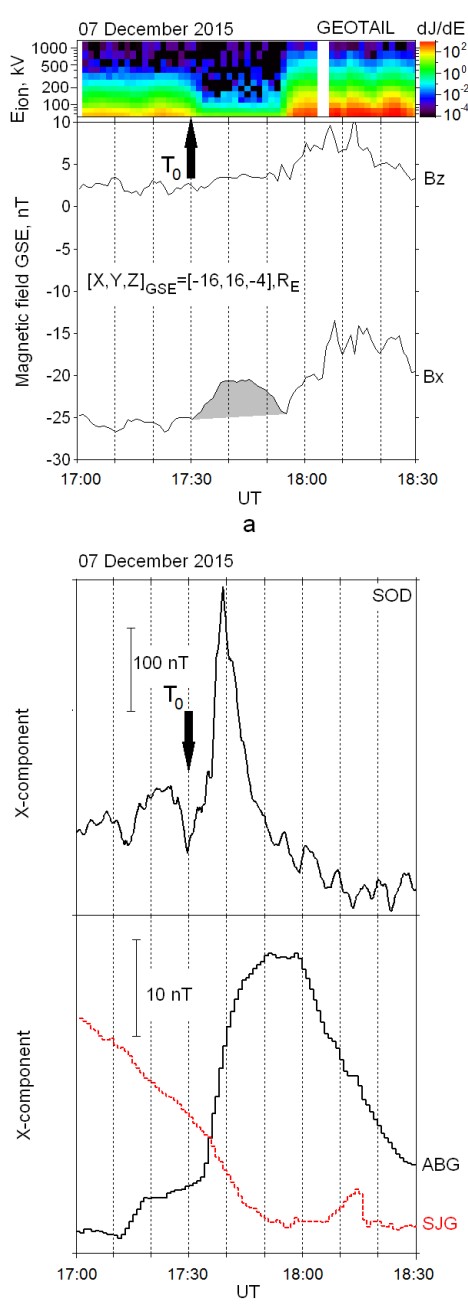

Figure 8. (a) spectrogram showing intensity variations of differential ion flux (*top panel*) and magnetic field at GEOTAIL (*bottom panel*); (b) variations of geomagnetic H-component at subauroral (SOD) and low-latitude (ABG, SGN) stations. Black arrow indicates the polar substorm onset time, $T_0$. Dayside variation is marked by red.



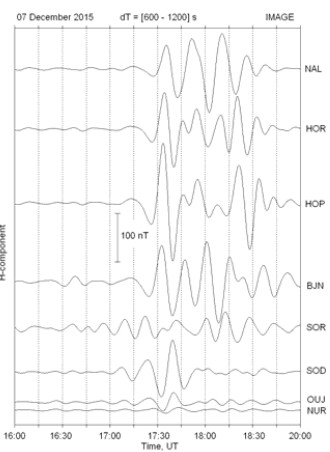

680                                                     a

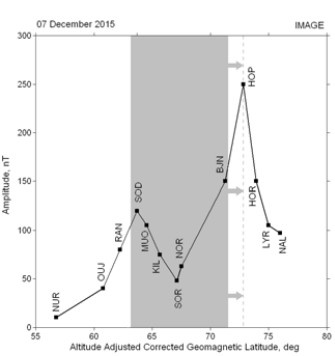

b

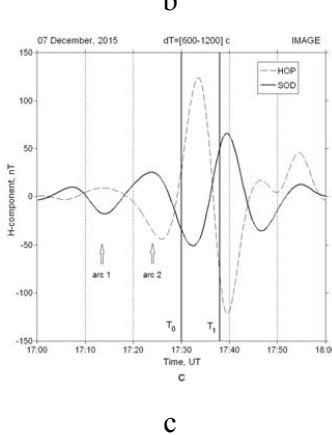

c

Figure 9. Wave "portrait" of polar substorms: (a) variations of H-component in a band 15± 5 min along meridian, the
presumable width of auroral oval is indicated with gray; (b) latitudinal distribution of pulsation intensity; (c) out-of-
phase variations at stations SOD and HOP where latitudinal distribution of pulsation intensity has maxima. Open arrows
indicate time of enhancement of pre-breakup arcs. $T_0$ and $T_1$ are the times of onset and torch formation, respectively.