# Peer review of "Polar substorm on 07 December 2015: pre-onset phenomena and features of auroral breakup"

_Annales Geophysicae, 2019_

## Referee Comment (RC1) · Anonymous Referee #1 · 8 Oct 2019

In this manuscript, a comprehensive analysis of a moderate "polar substorm" utilising ground-based observations combined with multi-spacecraft data, is presented. Their analysis of the substorm signatures are throughly described and easy to follow.

Except for the excellent description of the event, the authors also suggest that "oscillations" in the IMF Bz observed prior to the event might play a role in the substorm evolution. The oscillations in the IMF Bz are separated by approximately 15 minutes. The authors observe deflection on ground-based magnetometers, tailward flow in the polar cap (SuperDARN), and optical signatures matching the periodicity. This enhancement is also observed by ESR as a convecting density patch, which also signifies dayside

reconnection. It is also referred to earlier works by the author, where they suggested that such quasi-sinusoidal variations in IMF Bz could trigger substorms. It is then speculated that the periodicity of 15 minutes in the dayside reconnection will excite global oscillations with the same period in the magnetosphere, which in turn could trigger reconnection through some instabilities. While it is not discussed why such instabilities would be susceptible to oscillations with this specific periodicity, or how these oscillations are transmitted from polar regions into the plasma sheet, one can not argue with the author' speculations.

Below are some comments and suggestion that the authors may chose to include or exclude: L15: "Some later" -> "Later, " L27: "The question is solving on the base of satellite observations" , rewrite L53: Define WTS L66: Check reference (missing L) L119: per second L151: relative to L157: "Thus, by the moment .." -> "At T0, BJN ...." L207: "IMF deflections through 15 minutes" - rewrite L218: "suppose" -> "speculate" L253: "With taking" -> "Taking"

Also suggest to export figure 2 to a scalable format, also some of the labels are too small.

---

## Referee Comment (RC2) · Anonymous Referee #2 · 20 Dec 2019

Reviewer report on " Polar substorm on 07 December 2015: pre-onset phenomena and features of auroral breakup" by Safargaleev et al.

Summary: The paper discusses in great detail one substorm event that took place in the evening sector at high latitudes. Data from several instruments have been utilized. The most interesting result is presented in Figure 9, which shows 15-min magnetic oscillations at ground stations from OUJ to NAL, spanning roughly 13 deg in latitude. Obviously some auroral behavior show the 15-min periodicity, too. However, the paper contains a lot of unclear reasoning, in specific when the solar wind driving and possible dayside reconnection is discussed. The authors overinterpret the data, claiming that

two southward turnings of IMF Bz, separated roughly by 15 min, make IMF behaviour periodic and cause PERIODIC reconnection, which excites cavity oscillations in the magnetosphere. In addition, several timings must be checked. Therefore, the paper needs a major revision. However, I wish that finally the paper can be published because of these 15-min magnetic oscillations on the ground lasting at least for one hour and probably affecting substorm dynamics.

Major comments:

Section "Introduction" is focused on publications from 1990's and older, only some recent Russian papers are referred to. Why poleward boundary intensifications (PBIs) are not discussed here?

Section 3.1: Why AE index is not shown anywhere? It would help to put the event into global context, since the local magnetic time in Scandinavia is evening and not close to magnetic midnight.

l. 170: "The substorm was preceded by two negative variations in the H-component at KIL and SOD with repetition period of about 15 minutes (see Fig.2a middle panel)." (1) Give the times in the text. (2) If two events are separated by 15 min, this shouldn't be called repetition period.

Discussion of SuperDarn data in Section 3.2. is deficient. "...enhancement of the plasma flow in polar cap started at 17:04 UT, reached maximum at 17:08-17:10 UT (diagram d in Fig.4) and lasted until T0. One more flow enhancement took place at 16:52 UT, i.e. 15 minutes before the first one". If the intention is to make the readers confirmed that 15-min periodicity exists in SuperDarn data, then there should be either time series of velocities or all the panels, not just a few selected ones. Furthermore, it is not explained if the vectors represent l-o-s velocities or mapped velocities. In addition, typically IMF data is shown before discussing ionospheric responses. Now IMF data comes only in Figure 6.
l. 192: "The increase of F-region electron density at about T0 looks like a signature of the polar patch associated with the reconnected flux tubes drifting across the polar cap from the cusp to the magnetotail". – If the polar cap patch is formed on the dayside, near cusp, it takes a long time for this patch to drift over the polar cap to ESR. Please make that estimate.

l. 204: "Thus, the southward turning of IMF Bz could reach the magnetopause 20 min after registration onboard THC, and the ionospheric convection is expected to respond in $\sim$ 20 min after that (Hairson and Heelis, 1995)." – 20 min sounds a long time. Previous estimates of a global response have ranged from just seconds [Ridley et al., 1998] to 10 –15 min [e.g. Cowley and Lockwood, 1992]. However, if we use these numbers, they amount to 40-min delay, and then IMF variations at 16:15 and 16:30 UT correspond to 16:55 and 17:10 UT on the ground. I didn't see these numbers used in later discussion.

L. 294: "on the observation of the 15-minutes periodicity." Some features are obviously separated by 15 min, some 12 min.

Section 4.1, Figure 5. While the apparent vortices in equivalent current (which may be artefacts of data analysis in regions where they are no magnetometers) may only tentatively be associated with up- or downward FACs, why not the AMPERE data shown in Figure 5c is not discussed here?

l. 412: "The convection enhancements were caused by negative deviations of IMF Bz component" – This needs more convincing discussion in the paper, see my comments above.

l. 413: "Two weak variations in H-component might be the ground signature of global oscillations of the magnetospheric cavity excited by periodic erosion of the dayside magnetopause in the course of periodic reconnection" – Very unclear and hypothetical claim. Firstly, high- or low-latitude H-component? Secondly, I have not found any evidence in the data of PERIODIC reconnection. Two southward turnings of IMF Bz,

separated roughly by 15 min, doesn't make the IMF behaviour periodic. The claim is repeated in Conlusions, on l. 436.

l. 447: "The onset was accompanied by disruption of the dawn-to-dusk current in the plasma sheet around (X, Y) $\sim$ (-16, 16) RE" – With one single satellite showing an increase in the absolute value of the Bx component, one can only conclude that dipolarization has taken place, but it is not possible to pinpoint the location of current sheet disruption.

Minor comments:

l. 27: "The question is solving on the base of satellite observations" – Difficult to understand

l. 59: "Multiple onsets occur often." – Give a reference.

l. 78: "Samsonov et al. (2017) showed that the typical time for a southward interplanetary magnetic field turning to propagate across the magnetosheath is 14 min." – Dayside magnetosheath to subsolar magnetopause?

l. 103: "Fourth, we present GEOTAIL satellite data to investigate what process in plasma sheet – current disruption or neutral line formation – is responsible for the substorm onset (section 4.2)." – One satellite cannot give this information (was aim of Themis multi-satellite mission).

l. 111 BJN coordinates and elsewhere: Please specify if you use geographic or geomagnetic coordinates. Geomagnetic should be used.

l. 115 "footprints of localized downward (upward) field-aligned current (FAC) are manifested by quasi-circular clockwise (counterclockwise) equivalent current vortices around location of the upward (downward) FAC (e.g. Palin et al., 2016)." - This is a hypothesis and only valid for certain assumptions.

l. 201: "Assuming the nose of the bow shock at 14 RE" – Where is this estimate based

on?

l. 235: "At this moment the structure was stretched approximately along geomagnetic meridian and had dimension of 170x170 km." – A bit unclear description.

l. 260: "decrease in Bx component (indicated by gray shadow) while" - Actually the figure shows increase of Bx. However, the absolute value is decreased.

l. 265-267: Clarify the discussion, and make clear when dayside and when nightside low-latitude H is referred to.

l. 356: Spell out IPCL

————————————————————

---

## Author Comment (AC1) · 22 Jan 2020

Reply to Referee #1 interactive comments on "Polar substorm on 7 December 2015: pre-onset phenomena and features of auroral breakup" by Vladimir V. Safargaleev et al.

. . .It is then speculated that the periodicity of 15 minutes in the dayside reconnection will excite global oscillations with the same period in the magnetosphere, which in turn could trigger reconnection through some instabilities. While it is not discussed why such instabilities would be susceptible to oscillations with this specific periodicity, or how these oscillations are transmitted from polar regions into the plasma sheet, one

can not argue with the author' speculations.

- We indicate some instabilities as not the main but "one more possible" reason for substorm triggering (L425-428). In the cited papers by Roux et al. (1991) and Golovchanskaya et al. (2015) the conclusion regarding the substorm initiation via interchange or ballooning instabilities, respectively, was supported by satellite observations. Such satellite data are not available for the case considered in the present paper. In this respect we agree with Referee #1 comment that "one can not argue with the author' speculations" and remove the sentence from the text (L. 425-428).

Below are some comments and suggestion that the authors may chose to include or exclude: L15: "Some later" -> "Later, " L27: "The question is solving on the base of satellite observations" , rewrite L53: Define WTS L66: Check reference (missing L) L119: per second L151: relative to L157: "Thus, by the moment .." -> "At T0, BJN ...."L207: "IMF deflections through 15 minutes" - rewrite L218: "suppose" -> "speculate" L253: "With taking" -> "Taking" Also suggest to export figure 2 to a scalable format, also some of the labels are too small.

- We have considered these comments and made corresponding changes.

---

## Author Comment (AC2) · 22 Jan 2020

Reply to Referee #2 interactive comments on "Polar substorm on 7 December 2015: pre-onset phenomena and features of auroral breakup" by Vladimir V. Safargaleev et al.

Major comments:

Section "Introduction" is focused on publications from 1990's and older, only some recent Russian papers are referred to. Why poleward boundary intensifications (PBIs) are not discussed here?

[Figure]

- We added short comments on PBIs in Introduction and two papers in References. See also reply to Minor comments, L.27

Section 3.1: Why AE index is not shown anywhere? It would help to put the event into global context, since the local magnetic time in Scandinavia is evening and not close to magnetic midnight.

- We showed AE index in Fig.2 and added short comments in section 3.1 General overview...

l. 170: "The substorm was preceded by two negative variations in the H-component at KIL and SOD with repetition period of about 15 minutes (see Fig.2a middle panel)." (1) Give the times in the text. (2) If two events are separated by 15 min, this shouldn't be called repetition period.

- We would like to indicate the interval in the figure. Text is changed to: The substorm was preceded by two negative bays in the H-component at KIL and SOD at separation of about 15 minutes (interval is indicated with gray in Fig.2a). This variation is similar to a sinusoid and for brevity, hereinafter, we will use the term "repetition period" for the interval between two consecutive extremes (maxima or minima). These negative declinations . . ..

Discussion of SuperDarn data in Section 3.2. is deficient. "...enhancement of the plasma flow in polar cap started at 17:04 UT, reached maximum at 17:08-17:10 UT(diagram d in Fig.4) and lasted until T0. One more flow enhancement took place at 16:52 UT, i.e. 15 minutes before the first one". If the intention is to make the readers confirmed that 15-min periodicity exists in SuperDarn data, then there should be either time series of velocities or all the panels, not just a few selected ones. Furthermore, it is not explained if the vectors represent l-o-s velocities or mapped velocities. In addition, typically IMF data is shown before discussing ionospheric responses. Now IMF data comes only in Figure 6.

- We add panels in the figure. Now it shows the flow evolution at 2 min resolution. Vectors represent gridded line-of-sight velocities. The appropriate comment is added to figure caption. In our research we moved "from ground to solar wind", i.e. we looked for the variations in IMF which might cause convection enhancements by taking into account both the time delay and presumable shape of variation – two negative excursions at ~15-min separation. For this reason, IMF data are discussed and shown in the text after discussing the plasma flow variations in the ionosphere.

l. 192: "The increase of F-region electron density at about T0 looks like a signature of the polar patch associated with the reconnected flux tubes drifting across the polar cap from the cusp to the magnetotail". – If the polar cap patch is formed on the dayside, near cusp, it takes a long time for this patch to drift over the polar cap to ESR. Please make that estimate.

- Rewritten as following: Assuming that the patch was originated in the cusp region at the moment of first flow enhancement, one get the patch propagation time from the cusp to ESR beam to be ~ 40 min. Buchau et al. (1983) showed that patches drift antisunward with the background plasma flow (~1000 m/s that gives SuperDARN for the case considered). Thus, the distance between patch origin and place of patch detection is about 2500 km that corresponds approximately to the distance between statistical cusp position and the ESR beam.

l. 204: "Thus, the southward turning of IMF Bz could reach the magnetopause 20 min after registration onboard THC, and the ionospheric convection is expected to respond in _ 20 min after that (Hairson and Heelis, 1995)." – 20 min sounds a long time. Previous estimates of a global response have ranged from just seconds [Ridley et al., 1998] to 10 –15 min [e.g. Cowley and Lockwood, 1992]. However, if we use these numbers, they amount to 40-min delay, and then IMF variations at 16:15 and 16:30 UT correspond to 16:55 and 17:10 UT on the ground. I didn't see these numbers used in later discussion.

- As we mentioned in section 1.2 "The magnetospheric response time to the variation in the solar wind can vary from a few minutes to several hours." It is difficult to say what value should be taken for particular situation. Ridley et al. (1998) average estimation is up to 16.4 min. This estimation as well as estimation by Cowley and Lockwood (1992) is not dramatically smaller than 20 min estimation by Hairson and Heels (1998). In our research we moved "from ground to solar wind", i.e. we looked for the variations in IMF which might cause convection enhancements by taking into account both the time delay and presumable shape of variation – two negative excursions at ∼15-min separation. That is why we used the numbers in previous discussion, only.

L. 294: "on the observation of the 15-minutes periodicity." Some features are obviously separated by 15 min, some 12 min.

- We added the following remark in Section 5.3 The estimation of period depends on a number of factors, such as data resolution, subjectivism in the choice of the way of estimation (e.g. when we estimated repetition period of convection enhancements in polar cap and auroral activity over SOD), uncertainty in definition of the moment of max /min variations (e.g. when we estimated period as interval between two consecutive maximal declinations in H and Bz components), etc. So, it really is a period of 15±2 minutes, i.e. "close to 15 min" period. Thus, the term "15 min periodicity" is general and does not mean an exact value.

Section 4.1, Figure 5. While the apparent vortices in equivalent current (which may be artefacts of data analysis in regions where they are no magnetometers) may only tentatively be associated with up- or downward FACs, why not the AMPERE data shown in Figure 5c is not discussed here?

- Figure 5 shows ESR data. Distribution of equivalent currents is shown in Fig.7b and AMPERE data are shown at the same figure. See, as well, our comments to Referee's minor comments, L.115. It was important for us to define the location of presumable footprints of FACs relatively auroras. It was easy to do by the use IMAGE data because

AMPERE gives only a general view of FACs position.

l. 412: "The convection enhancements were caused by negative deviations of IMF Bz component" – This needs more convincing discussion in the paper, see my comments above.

- Enhancement of antisunward ionospheric convection across the polar cap is traditionally connected with reconnection initiated by southward turning of IMF Bz component (e.g. Ruohoniemi and Greenwald, 1998).

l. 413: "Two weak variations in H-component might be the ground signature of global oscillations of the magnetospheric cavity excited by periodic erosion of the dayside magnetopause in the course of periodic reconnection" – Very unclear and hypothetical claim. Firstly, high- or low-latitude H-component? Secondly, I have not found any evidence in the data of PERIODIC reconnection. Two southward turnings of IMF Bz, separated roughly by 15 min, doesn't make the IMF behaviour periodic. The claim is repeated in Conlusions, on l. 436.

- Rewritten as following: Two weak variations in H-component at KIL and SOD might be the ground signature of global oscillations of the magnetospheric cavity (see Fig.9). The oscillations might be excited by periodic erosion of the dayside magnetopause in the course of periodic reconnection (e.g. Agapitov et al., 2009). The conclusion regarding periodic reconnection is based on periodic enhancement of plasma velocity in the polar cap (see section 3.2).

l. 447: "The onset was accompanied by disruption of the dawn-to-dusk current in the plasma sheet around (X, Y) _ (-16, 16) RE" – With one single satellite showing an increase in the absolute value of the Bx component, one can only conclude that dipolarization has taken place, but it is not possible to pinpoint the location of current sheet disruption.

- Our comments. We would like to remind that in the early years the magnetospheric

studies were based on one single satellite observations. In particularity, the concept of dipolarization was inferred from North-South field behavior (e.g. Miashyta et al., 2000). In the case considered in our paper, GEOTAIL was located slightly southward of the current sheet. The hypothesis is supported not only by the coordinates of satellite position but also by large Bx and near-zero Bz components. At 17:30 UT CEOTAIL showed the DECREASE in the absolute value of Bx (from 25 to 20 nT) while Bz stayed almost constant. This means the decrease of dawn-to-dusk current in the current sheet. We corrected the text as the following: We conclude this from data of the GEOTAIL satellite showing the reduction in the absolute value of the Bx component (e.g. Lui et al., 1992) and dropout of high-energy electrons, enhancement of the westward electrojet and the large positive variation in H-component at low latitudes. In accordance with Lui (1996), current disruption activity is limited both radially and azimuthally to -1 RE. Since the GEOTAIL turned to be sensitive to the changes in Bx and electron flux and was magnetically conjugated with changing electrojet, we suggest that current decrease/disruption took place in the satellite vicinity.

Minor comments: l. 27: "The question is solving on the base of satellite observations" – Difficult to understand

- In the distant magnetotail, the direct comparison of satellite measurements and ground data is hindered by the low accuracy of mapping of magnetospheric processes to the ionosphere because of the complex shape of geomagnetic field lines. In particular, the causal link between the formation of so-called auroral "poleward boundary intensifications" (PBIs) and distant reconnection (e.g. Lyons et al., 1999) is very difficult to test. Note, that some kind of PBIs is regarded as substorm onset trigger (Nishimura et al., 2015). To solve the above problem one needs either appropriate modification of geomagnetic field model. . .

l. 59: "Multiple onsets occur often." – Give a reference.

- Baker et al. (1996) noticed that multiple onsets occur often.

l. 78: "Samsonov et al. (2017) showed that the typical time for a southward interplanetary magnetic field turning to propagate across the magnetosheath is 14 min." – Dayside magnetosheath to subsolar magnetopause?

- Yes. We indicate this in the text.

l. 103: "Fourth, we present GEOTAIL satellite data to investigate what process in plasma sheet – current disruption or neutral line formation – is responsible for the substorm onset (section 4.2)." – One satellite cannot give this information (was aim of Themis multi-satellite mission).

- We would like to remind again that in the early years the magnetospheric studies (including substorm disturbances in the magnetotail plasma sheet) were based on one single satellite observations. Nevertheless, we rewrite the sentence as following. "Fourth, we present GEOTAIL satellite data to show that in the case considered the current disruption in plasma sheet is more probable reason for the substorm onset than the neutral line formation (section 4.2)."

l. 111 BJN coordinates and elsewhere: Please specify if you use geographic or geomagnetic coordinates. Geomagnetic should be used.

- We add geomagnetic coordinates of the stations in the cases if only geographic ones are indicated in the text. We would like to retain geographic coordinates too because coordinates on the map in fig.1 and 7b, as well as in fig.3b are geographic.

l. 115 "footprints of localized downward (upward) field-aligned current (FAC) are manifested by quasi-circular clockwise (counterclockwise) equivalent current vortices around location of the upward (downward) FAC (e.g. Palin et al., 2016)." - This is a hypothesis and only valid for certain assumptions.

- Yes. This is assumption, but it is widely used assumption. Additionally, we presented AMPERE satellite data showing the same location of the field-aligned currents as defined from IMAGE magnetometer data (Sections 2 and 5.2).

[Figure]

l. 201: "Assuming the nose of the bow shock at 14 RE" – Where is this estimate based on?

- This estimation is taken from the 4-D Orbit Viewer tool mentioned in the text (L.198).

l. 235: "At this moment the structure was stretched approximately along geomagnetic meridian and had dimension of 170x170 km." – A bit unclear description.

- Changed to: At this moment the structure was oriented approximately along geomagnetic meridian and had dimension of 170x170 km

l. 260: "decrease in Bx component (indicated by gray shadow) while" - Actually the figure shows increase of Bx. However, the absolute value is decreased.

- We rewrite as "...decrease in absolute value of Bx component (indicated by gray shadow) while..."

l. 265-267: Clarify the discussion, and make clear when dayside and when nightside low-latitude H is referred to.

- Rewritten as following: The increase of H-component at low latitudes in all MLT sectors is traditionally connected with the enhancement of solar wind dynamic pressure which is not occurred in the present case (see Fig.7b and variation in H-component at the dayside station San Juan, SJG, 18.1°N, 293.8°E in Fig.8b). In accordance with Maltsev et al. (1996) and Huang et al. (2004), the cross-tail magnetospheric current also contributes to the Dst variation, i.e., to the H-component at equatorial latitudes. Hence, the magnetic effect of the decrease or disruption of this current in the nightside magnetosphere will be manifested as the H-component increase at low latitude stations located, as well, on the nightside.

l. 356: Spell out IPCL

- ... IPCL (irregular pulsations, continuous, long)...

---

## Author Response (AR2)

**Reply to Referee comments on "Polar substorm on 7 December 2015: pre-onset phenomena and features of auroral breakup" by Vladimir V. Safargaleev et al.**
(Referee comments are bold, our corrections are italic. Corrections in the text of paper are yellow).

MAJOR COMMENTS

**As noted previously by reviewer #2, it is strange to talk about periodic structures and periodic reconnection etc. when there are only 2 instances. I recommend that you reduce talk of periodicity and instead talk about 2 structures separated by ~15 minutes. I have made several detailed comments about this below (see minor comments), but have surely missed several points. Especially in Lines 368-372 you should remind the reader that your "periodicity" is based on 2 instances.**
We have reduced talk of periodicity indicated in minor comments as it was recommended and add the following sentence in Lines 371 - 372:
*We remind the reader that by 15-min periodicity of a parameter we mean two its changes, following one after the other with an interval of 15 minutes.*

**L70: I'm not sure if Mishin et al. (2001) discussed "quasi-sinusoidal" IMF BZ variations as a necessary condition for triggering a polar substorm. They did discuss the 2-step sequence of the substorm, but I don't think quasi-sinusoidal variations were discussed. Also, the examples in Mishin et al. (2001) seem to show a large variation in timing, not always ~15 minutes. Could you clarify this?**
Yes, examples in Mishin et al. (2001) show a larger variation in timing. We have cited this paper to show that sinusoid-like variations in IMF Bz (i.e. gradual change, at first, to negative and then to positive values) were discussed earlier in the context of substorm initiation. To avoid the misunderstanding we rewrite this part of Introduction as follows (L.69-78):

*Russell (2000) suggested that double substorm onsets can be caused by a temporal deflection of northward IMF to southward. In the review by Baker et al. (1996) it was noted a class of substorms that were triggered by positive changes in Bz after it turned to south. Mishin et al. (2001) showed by the superimposed epoch analysis that substorm associated Bz variation is a gradual change at first to negative and then to positive values and looks like a fragment of sinusoid. As a rule, the above fluctuations are easy identified in IMF data due to large amplitude and time scale or inferred by statistics. Recently, Safargaleev et al. (2018) proposed that the polar substorm might be initiated by the less prominent sinusoid-like variation in IMF Bz component with period ~ 15 min detected in the solar wind several tens minutes prior onset. To associate substorm onset with such kind of IMF variations one needs careful estimating of the time delay between the arrival of IMF irregularity to the magnetopause and the beginning of the substorm.*

**L120-122: As mentioned by reviewer #2, the relation between equivalent current and FAC is approximate. Also Palin et al write "...FAC can sometimes be identified by a quasi- circular clockwise (counterclockwise) equivalent current vortex...". More specifically, in order to associate curl of the equivalent current with FAC, you need to assume that conductance gradients are parallel to the electric field, see e.g. section 2 in Amm (2002) for details. Therefore I'd recommend that you change in line 120 "are manifested by" --> "can often be associated with" and add reference to Amm et al. (2002).**
Corrected. L.125-127

**L259-260: The vortices are in an area where there are few magnetometers. Uncertainty in the equivalent currents increases in these areas, and in my experience there may be spurious vortex-like structures over oceans, where there are no magnetometers. Therefore I'd suggest you to re-phrase the sentence and say something like "The vortices seen in the equivalent**

**current are consistent with downward FAC at the poleward side of the coiling structure and upward FAC equatorward of it.".**
Corrected. L.266-267.

**L204: Do you mean that the polar patch caused the shift in the electrojet, or that they just happened to occur at the same time?**
The patches and displacement happened to occur at the same time.  For this reason, we suppose that the arrival of the reconnected flux tubes, which footprints are detected by EISCAT radar as the patches, to the lobe could be resulted in the lobe expansion. As a sequence, the polar cap expanded and auroral oval shifted toward the lower latitudes together with the electrojet.
The following comment is added in the text:
*Appearance of the polar patch in radar data and the equatorward shift of westward electojet (Fig.2b) happened to occur at the same time. Assuming the patch to be the footprint of one of the reconnected flux tubes, we suppose that the jet displacements could be a sequence of expansion of magnetospheric lobe caused by reconnected flux tubes, arriving from the dayside.  (L209-212)*

**L386-390: Is there a peak in the power spectra around 15 minutes, or did you just select the 0.8-1.7 mHz band based on earlier observations of structures separated by ~15 minutes? If the latter is true, then it would be good to check the power spectra. Looking at the magnetograms in Fig2 it seems that there could be stuff also at shorter periods.**
Yes, there is a peak in power spectrum at ~0.001 Hz (15 min) in variations at BJN station where the maximum of negative bay is achieved. We have demonstrated this by a new figure 9a and corresponding comments in the text (L393-398) :
To find15 minute variations using FFT, one need to analyse a long interval (not less than one hour) which include many different variations before and after onset. So, the power spectrum is an integral characteristic of the interval. From this point of view, the procedure of filtering that w used looks preferable.

MINOR COMMENTS

**L11: Repetition period --> time interval**
Corrected

**L31: PBIs is --> PBIs are & trigger --> triggers**
In the text "is" refers to "some kind". We are not sure that correction is needed.

**L46: Pulkinen --> Pulkkinen**
Corrected

**L71: by the --> by a**
Corrected

**L86: Heikila --> Heikkila**
Corrected

**L94: onset so that such --> onset**, so such
Corrected

**L108: 15 min oscillations --> 2 structures separated by 15 min**
Corrected

**L143: event was --> event took place & No a --> No**

Corrected

**L185: As it was mentioned --> As mentioned**
Corrected

**L193: So that, the increase --> This increase**
Corrected

**section 3.2: Check references to panels in Fig 4, some of them may refer to the old version.**
Corrected

**L201: that gives --> estimated from**
Corrected

**L216: which --> whose**
Corrected

**L239: repetition of variation --> interval between the two negative bays**
Corrected

**L276-284: I had to read this couple times to get the point. I recommend you re-write it, for example: "The bottom panel in Fig.8b shows variation in the H magnetic field component at the low-latitude stations Alibag (ABG, 18.5°N, 72.9°E; geomagnetic latitude 11.65°N) located near midnight and at the dayside station San Juan (SJG, 18.1°N, 293.8°E; geomagnetic latitude 28.79°N). The increase of H-component at low latitudes in all MLT sectors is traditionally connected with the enhancement of solar wind dynamic pressurem, while decrease or disruption of the cross-tail magnetospheric current contributes to the Dst variation mainly on the nightside (Maltsev et al. 1996; Huang et al. 2004). Thus the very different magnetic field behaviour seen at ABG and SJG support current disruption of the cross-tail current."**
Corrected as referee recommended

**L304: of the same periodicity --> with the same time separation**
Corrected

**L306: 15-minutes periodicity --> 15-minute separation.**
Corrected

**L330: at latitudes --> at magnetic latitudes**
Corrected

**L363: 15-min periodicity --> 15-min time separation**
Corrected

**L372: period --> interval**
In the context, the term "period" refers to classification of geomagnetic pulsation. We think that correction is not needed.

**L387: 0.8 - 1.7**
Corrected

**L401: periodic --> two**

Corrected

**L417 and elsewhere: use consistently e**ither arcX or arc X
Corrected

**L453: repetition period --> time sep**aration
Corrected

**L454: periodic reconnection --> two bursts of reconnection**
Corrected

**L457: periodic erosion --> repeated erosion**
Corrected

**L466: In accordance with --> According to**
Corrected

**L467: turned out to be** sensitive to the --> detected
Corrected

**L469 satellite --> satellite's**
Corrected

**Fig2: Numbers in the panel showing AE index are too small. Simiarly the white text in the high resolution SOD and BAB keograms is difficult to see.**
Corrected

**Fig 3b: Is the colorbar supposed to have another set of numbers on the left side, or why there is text "electrons" and "ions" in blue and red and numbers only in red?**
Corrected

**Fig 5 caption: arives --> arriving**
Corrected

**Fig 8 caption: SGN --> SJG**
Corrected